# Delving Deep into Sim2Real Transformation: Maximizing Impact of Synthetic Data in Training

## Abstract

Synthetic data has the distinct advantage of building a large-scale labeled dataset for almost free. Still, it should be carefully integrated into learning; otherwise, the expected performance gains are difficult to achieve. The biggest hurdle for synthetic data to achieve increased training performance is the domain gap with the (real) test data. As a common solution to deal with the domain gap, the *sim2real transformation* is used, and its quality is affected by three factors: i) the real data serving as a reference when calculating the domain gap, ii) the synthetic data chosen to avoid the transformation quality degradation, and iii) the synthetic data pool from which the synthetic data is selected. In this paper, we investigate the impact of these factors on maximizing the effectiveness of synthetic data in training in terms of improving learning performance and acquiring domain generalization ability–two main benefits expected of using synthetic data. As an evaluation metric for the second benefit, we introduce a method for measuring the distribution gap between two datasets, which is derived as the normalized sum of the Mahalanobis distances of all test data. As a result, we have discovered several important findings that have never been investigated or have been used previously without accurate understanding. We expect that these findings can break the current trend of either naively using or being hesitant to use synthetic data in machine learning due to the lack of understanding, leading to more appropriate use in future research.

## 1 Introduction

As large-capacity models (*e.g.*, CNN, transformer) start to present significant impacts on a variety of machine learning problems, supplying sufficient amounts of data to train the models becomes an urgent issue. Accordingly, the demand for synthetic data rapidly grows due to its nearly zero cost in building large-scale labeled data. Many attempts to use synthetic data for training data augmentation have been made, but properly creating synthetic data relevant to given learning tasks remains a challenge. That is mainly because fully exploiting the inherent strengths of synthetic data requires an appropriate understanding of various properties inducing the domain gap compared to real data.

A common solution for bridging the domain gap is to transform the properties of synthetic data to enhance realism. In general, the sim2real transformers are trained on a source-to-target adaptation framework (*e.g.*, conditional GAN (Zhu et al., 2017; Hoffman et al., 2018; Shen et al., 2023a)), treating synthetic and real data as the source and target domains, respectively. However, the satisfactory quality of the sim2real transformation cannot be expected if the domain gap between the two sets is not manageable to overcome. One effective way to deal with this dilemma, where the sim2real transformation to handle the large domain gap is negatively affected by the large domain gap, is to use only a portion of synthetic data with a small domain gap

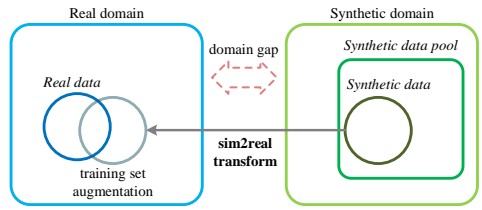

Figure 1: **Sim2real transformation mechanism.** Three datasets (real data, synthetic data, and the synthetic data pool) can influence the impact of synthetic data used in training.

with the real data. To take advantage of the diversity of synthetic data in training, data with relatively large domain gaps can also be selected at a lower rate. (Shen et al., 2023a) In summary, there are three factors that affect sim2real transformation quality (Fig. 1): i) the real data serving as a reference when measuring the domain gap, ii) the selected synthetic data used in training, and iii) the synthetic data pool from which a subset of synthetic data is selected. To maximize the impact of synthetic data in training while adequately addressing the domain gap, a thorough investigation into how the three factors play into optimally integrating synthetic data into learning is critical.

In this study, we evaluated the impact of synthetic data in a same-domain task where the training and test sets were built on the same domain, and a cross-domain task where the two sets were from different domains. With the two tasks, we aim to ensure the following two benefits of using synthetic data in training are realized: i) improving learning performance by training with hybrid sets of real and synthetic data, and ii) leading the model to acquire domain generalization ability to achieve satisfactory accuracy regardless of the dataset in a cross-domain task. To do that, we firstly evaluated the model performance. We also measured the distribution gap between the training and test sets in a cross-domain task with and without using synthetic data in training to specifically confirm the second benefit. As a measure of the distribution gap, we use the cross-entropy while representing the distributions of the two sets as a multivariate Gaussian distribution and a mixture of Delta distribution, respectively. We have shown that, theoretically, the distribution for a certain category of a training set used in detector training can be represented as a multivariate Gaussian distribution in the detector's representation space. In the end, the distribution gap can be derived as the normalized sum of the Mahalanobis distances (Mahalanobis, 1936) from the training set for each test data.

After carrying out a comprehensive study based on extensive experiments with the two measurements (detection accuracy and distribution gaps), we uncovered the following useful findings:

1) In a cross-domain task, using synthetic data helps to significantly reduce the distribution gaps of most of the test data but also unexpectedly produces considerably large distribution gaps for some outlier data.

2) To enhance the impact of synthetic data in training, it is important to increase the amount of not only synthetic data but also real data in both the same-domain and cross-domain tasks.

3) In a cross-domain task, improving the sim2real transformation quality of the synthetic data is more effective in leading the model to acquiring domain generalization ability than reducing the distribution gap between the training and test sets when achieving the two objectives together is not feasible.

4) The properties of the synthetic data pool (*i.e.*, the density and diversity of the synthetic data distribution in the feature space, and the domain gap with respect to the real data) also influence the enhancement of impact from using synthetic data in training.

In recent works, the above findings have not been carefully considered or have been used without accurate understanding. We provide empirical evidence verifying the findings through extensive experiments. We hope that our study can provide a clue for a breakthrough that can address the community's hesitant or improper use of synthetic data.

## 2 RELATED WORKS

**Measuring distribution gap between two datasets.** Measuring the differences in the properties of distinctive datasets is crucial for analyzing performance in cross-dataset tasks (*e.g.*, domain adaptation/generalization, sim2real transformation). Measurements depend on which property is focused on in the analysis. To measure the differences in *class conditional distributions* of two datasets, scatter (Ghifary et al., 2017), maximum mean discrepancy (MMD) (Yan et al., 2017; Li et al., 2018), high-order MMD (HoMM) (Chen et al., 2020), *etc*. are used. Statistical measures over the distances between samples of different datasets in the *feature space* are also considered to estimate the distribution gap of the datasets. Here, the feature space can be learned in a direction of preserving the properties of the synthetic data in the sim2real transformation (Zhu et al., 2017) or minimizing the feature distribution of two datasets through contrastive learning (Motiian et al., 2017; Yao et al., 2022) or knowledge distillation (Dobler et al., 2023). All the methods above are involved in training as a loss function for learning the dataset-invariant representation. On the other hand, we use the distribution gap measure to investigate its relationship with *post-training* performance.

**Exploring proper uses of synthetic data.** It is challenging to expect effectiveness in training with synthetic data without adequately addressing the domain gaps with real-world test sets. One category of leveraging synthetic data in training employs special processing to reduce domain gaps when generating synthetic data, *e.g.*, incorporating some real-world components (texture, background) (Peng et al., 2015; Wang et al., 2020; 2022; Dokania et al., 2022; Li et al., 2023b) and cloning real sets (Ros et al., 2016; Liu et al., 2023; Zeng et al., 2023). Synthetic data created by simply injecting noise (Li et al., 2023a), messiness (Wei et al., 2023), or simple-shape objects such as rain (Ba et al., 2022), is relatively free of the domain gap. There also exist other methods (Wu et al., 2022; Ni et al., 2023) to bridge the domain gap relying on recently emerging high-performance image generators (*e.g.*, CLIP (Radford et al., 2021), VQ-GAN (Esser et al., 2021)).

Unfortunately, the aforementioned methods do not provide a comprehensive solution for reducing the domain gap. Among more general solutions recently developed, some methods mitigate the domain gap rather than completely reducing it via creating *easily generalizable* feature embeddings instead of raw data (Su et al., 2022; Kumar et al., 2023), or adjusting the ratio with the real data during training (Ros et al., 2016; Richter et al., 2016; Lee et al., 2021). The recently introduced PTL (Shen et al., 2023a) is a method that iteratively selects subsets of synthetic data while accounting for domain gaps, resulting in significant performance gains in general detection tasks.

**Analyzing properties of synthetic datasets.** There are many studies that have analyzed synthetic data in various aspects, such as safety/reliability (Zendel et al., 2017; He et al., 2023), diversity (Gao et al., 2023; He et al., 2023), density/coverage (Naeem et al., 2020; Han et al., 2023), *etc*. The impact of using synthetic datasets has been analyzed according to the scalability (Liu et al., 2022; Sariyildiz et al., 2023) or variation factors used to build the dataset (Tang & Jia, 2023). Li et al. (2023c) observes accuracy in same-domain and cross-domain tasks in the Visual Question Answering (VQA) task to figure out the transfer capability of synthetic data. While the aforementioned work performed these analyzes on specific synthetic datasets, we have carried out more general and comprehensive analyses on various aspects.

## 3 METHODOLOGY

Our primary goal is to conduct a comprehensive study to find the environment that maximizes the two expected benefits of using synthetic data: i) improving performance, and ii) leading the model to acquire domain generalization ability. To fulfil this goal, in particular to ensure if the second benefit is realized, we first introduce how to theoretically measure the distribution gap between train and test sets in a cross-domain task. Then, we introduce a recently introduced method that provides a simple yet effective way to leverage synthetic images in training, *i.e.*, PTL (Shen et al., 2023a). PTL was remarkably better at providing detection accuracy and acquiring domain generalizability compared to other counterparts that also leverage synthetic images (*e.g.*, naive merge and pretrain-finetune). We found a strategy to reduce PTL's training time, crucial to completing large volumes of comprehensive experiments.

### 3.1 MEASURING DISTRIBUTION GAP

**Modeling the dataset with multivariate Gaussian distribution.** The distribution of a dataset for a specific category can be modeled as a multivariate Gaussian distribution in the representation space of a detector trained on the dataset if the following two conditions are satisfied: i) the detector takes the form of sigmoid-based outputs and ii) the representation space is built with the output of the penultimate layer of the detector.[1] Specifically, let $\mathbf{x} \in \mathcal{X}$ and $y = \{y_c\}_{c=1,\cdots,C} \in \mathcal{Y}$, $y_c \in \{0, 1\}$ be an input and its categorical label, respectively. Then, the representation for the category $c$ can be expressed as follows:

$$P(f(\mathbf{x})|y_c = 1) = \mathcal{N}(f(\mathbf{x})|\mu_c, \Sigma_c), \tag{1}$$

where $f(\cdot)$ denotes the output of the penultimate layer of the detector. $\mu_c$ and $\Sigma_c$ are the mean and the covariance (*i.e.*, two parameters defining the multivariate Gaussian distribution) of the representation for the category $c$, respectively.[2] These parameters can be computed empirically with the dataset.

---

[1]This modeling is proven in the supplementary material.

[2]Hereafter, since the target object is human only in this paper, we use terms without subscript $c$, meaning a specific category, throughout the paper.

**Distribution gap to the new dataset.** To measure the distribution gap between two datasets (*i.e.*, a reference dataset $\mathcal{D}_r$ and a new dataset $\mathcal{D}_{\text{new}}$), we used the cross-entropy, which statistically measures how a given distribution is different from the reference distribution. (i.e., $\mathcal{H}(P, Q) = -\int_{\mathcal{X}} p(\mathbf{x}) \ln q(\mathbf{x}) d\mathbf{x}$, where $p$ and $q$ denote the probability densities of two distributions $P$ and $Q$, respectively. Here, $Q$ is the reference distribution.) We regard the dataset where the representation space is built as the reference dataset and calculate the distribution gap from the reference dataset to the new dataset in the representation space.

As demonstrated in the previous section, the probability density of $\mathcal{D}_r$ can be expressed as a multivariate Gaussian distribution, as in eq 1. Since $\mathcal{D}_{\text{new}}$ is not involved in detector training, we regard the probability density of the dataset as a mixture model where each component indicating a single element of the dataset takes the form of a Dirac delta function, as follows:

$$p(\mathbf{x}) = \frac{1}{|\mathcal{D}_{\text{new}}|} \sum_{\mathbf{x}' \in \mathcal{D}_{\text{new}}} \delta(\mathbf{x} - \mathbf{x}'), \tag{2}$$

where $\delta(\mathbf{x})$ is a Dirac delta function whose value is zero everywhere except at $\mathbf{x} = 0$ and whose integral over $\mathcal{X}$ which is the entire space of $\mathbf{x}$ is one (*i.e.*, $\int_{\mathcal{X}} \mathbf{x} d\mathbf{x} = 1$).

Using the two probability densities of $P$ and $Q$ defined in eq 1 and 2, cross-entropy can be derived[3], as:

$$\mathcal{H}(P, Q) = \frac{1}{2|\mathcal{D}_{\text{new}}|} \sum_{\mathbf{x} \in \mathcal{D}_{\text{new}}} (f(\mathbf{x}) - \mu)^\top \Sigma^{-1} (f(\mathbf{x}) - \mu) + C, \tag{3}$$

where $C$ is a constant that is not affected by $\mathcal{D}_{\text{new}}$. Accordingly, to quantitatively compare distribution gaps of two new datasets with respect to the reference dataset, we define a distribution gap for the new dataset by removing $C$ from the cross-entropy in 3, as:

$$d(\mathcal{D}_{\text{new}} | \mu, \Sigma) = \frac{1}{2|\mathcal{D}_{\text{new}}|} \sum_{\mathbf{x} \in \mathcal{D}_{\text{new}}} (f(\mathbf{x}) - \mu)^\top \Sigma^{-1} (f(\mathbf{x}) - \mu). \tag{4}$$

As a result, the distribution gap measure takes the form of a normalized sum of the Mahalanobis distances Mahalanobis (1936) over all data in $\mathcal{D}_{\text{new}}$.

## 3.2 Leveraging Synthetic Images in Training

**Progressive Transformation Learning (PTL).** PTL gradually expands training data by repeating two steps: i) selecting a subset of synthetic data and ii) transforming the selected synthetic images to look more realistic. This progressive strategy is used to address quality degradation of the sim2real transformation that can occur due to the large domain gap between the real and the synthetic domains.

The subset of the synthetic set is constructed by selecting more synthetic images with a closer domain gap to the training set. The *sim2real transformer* is trained via a conditional GAN (specifically, CycleGAN (Zhu et al., 2017)) to transform selected synthetic images to have the visual properties of the current training set. Note that two training processes for the detector and the sim2real transformer are involved in the PTL process for each iteration.

**PTL training time curtailment.** The biggest bottleneck when conducting a comprehensive study with PTL is the lengthy training time (*e.g.*, 10 and 16 hours for PTL training under the Vis-20/Vis-50 [4] setups, respectively). Sim2real transformer training takes up the largest portion of PTL training time, followed by detector training. Originally, these two training processes start from scratch for every PTL iteration because the training set changes with every PTL iteration. Instead of this time-consuming training approach, we consider the *tuning-from-previous-iteration* strategy, where the model to be trained is initialized from the model learned in the previous PTL iteration, with fewer training iterations.

Table 1 shows the change in training time and accuracy with this time-curtailing strategy on the Vis-20/50 setups. When using this strategy for sim2real transformer training, it was effective as the

---

[3]This derivation can be found in the supplementary material.

[4]We refer to the setting using $N$ images of the VisDrone dataset as a real training set as 'Vis-N' throughout all experiments, e.g., Vis-20.

time was significantly reduced (at least $\times 0.65$) without loss of accuracy. On the other hand, applying this strategy in training the detector (with the sim2real transformer training) has a negative impact as accuracy is significantly reduced but time curtailment is not as great as that achieved with the sim2real transformer solely. Based on this comparison, we used the *tune-from-previous-iteration* strategy in training the sim2real transformer only throughout the following experiments.

Table 1: **Training time curtailment via 'tuning-from-previous-iteration' strategy.** $f_t$ and $f_d$ represent the sim2real transformer and the detector, respectively. The training time and the accuracy are measured with wall-clock time in hours and AP@[.5:.95], respectively. The number in parentheses in 'time' indicates the relative time compared to the original PTL training.

| from-prev-iter | Vis-20 | | | | Vis-50 | | | |
|---|---|---|---|---|---|---|---|---|
| | time | Vis | Oku | ICG | time | Vis | Oku | ICG |
| Original | 10 | 1.94 | 7.45 | 7.22 | 16 | 2.85 | 11.46 | 7.27 |
| $f_t$ | 6.5 ($\times$0.65) | 1.95 | 7.01 | 8.93 | 9 ($\times$0.56) | 2.78 | 11.52 | 9.90 |
| $f_t$ & $f_d$ | 5.5 ($\times$0.55) | 1.61 | 6.03 | 4.71 | 7.5 ($\times$0.47) | 2.43 | 9.81 | 6.78 |

## 4 EXPERIMENTAL SETTINGS

**Task and dataset.** Our comprehensive study is conducted on human detection in UAV-view images. In a UAV-view image, where a person's appearance becomes very diverse, the need for synthetic data is more pressing. In addition, we also use $N$-shot detection tasks, where a limited number of $N$ images, are used for training, and cross-domain detection tasks, where the same domain images are not available in training.

We use five datasets built for UAV-view human detection for real datasets: VisDrone (Zhu et al., 2022), Okutama-Action (Barekatain et al., 2017), ICG (ICG), HERIDAL (Božić-Štulić et al., 2019), and SARD (Sambolek & Ivasic-Kos, 2021). VisDrone is used as a training set, and all five datasets are used as test sets. For a synthetic data pool, we use the Archangel-Synthetic (Shen et al., 2023b).

Our criteria for selecting a task are i) whether the task has a high demand for synthetic data, and ii) whether a synthetic dataset exists that can be used for comprehensive study for sim2real transformation. UAV-view human detection, $N$-shot detection, and cross-domain detection are tasks required in a problem space where real data is extremely scarce; thus synthetic data is in high demand. And the recently introduced Archangel-Synthetic dataset is suitable for conducting our comprehensive study in these tasks because it is large-scale and provides meta-data about the rendering parameters used to build the dataset.

**Evaluation metrics.** We use MS COCO style AP@.5 and AP@[.5:.95] as evaluation metrics in our study. Due to space limitations, only AP@[.5:.95] is reported in the main manuscript while AP@.5 values are additionally reported in the supplementary material. We perform three runs and report the average value to address potential random effects in the $N$-shot detection task.

## 5 RESULTS AND ANALYSIS

### 5.1 A STUDY ON THE IMPACT OF REAL DATA

For the first study, we explore the scalability behavior of real data regarding the two impacts of using synthetic data in training: i) increasing detection accuracy and ii) reducing the distribution gaps. Specifically, these two aspects are compared among four cases using real data with different quantities (*i.e.*, 20, 50, 100, and 200).

**Analysis in terms of accuracy.** In same-domain tasks (Fig. 2a), detection accuracy unsurprisingly increased proportionally with the size of the real dataset, regardless of whether or not synthetic data is used. The use of synthetic data consistently increases accuracy irrespective of the size of the real data set. Interestingly, adopting a larger real dataset yields better accuracy even in most cross-domain tasks (Fig. 2b). These trends indicate that it is essential to use synthetic data with real data, including cross-domain tasks.

Using a large number of real images (*i.e.*, 200), on the other hand, results in little increase or adversely affects accuracy compared to using fewer images. The rationale behind this notable observation can be accounted for with Fig. 2c, which illustrates the accuracy ratio of a cross-domain task to a same-domain task. Here, the two tasks use different training sets (for the cross-domain task, we use VisDrone as the training set.) but are evaluated on the same test set. When the number of real images

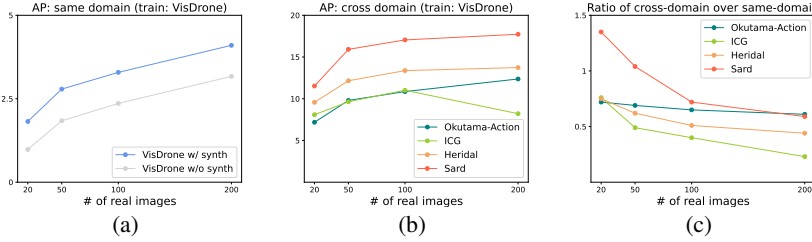

(a)                             (b)                             (c)

Figure 2: **Accuracy with the size of real dataset.** (b) and (c) show the accuracy when synthetic images are used in training.

Table 2: **Distribution gaps of various datasets** from VisDrone. The better accuracy between using and not using synthetic images is shown in bold.

| setup | w/ synth | Okutama 50% | Okutama all | ICG 50% | ICG all | HERIDAL 50% | HERIDAL all | SARD 50% | SARD all |
|---|---|---|---|---|---|---|---|---|---|
| Vis-20 | | 90.6 | 4,627.1 | 63.0 | 2,372.9 | 151.0 | 6,524.2 | 132.2 | 5,083.5 |
| | ✓ | **35.1** | **487.9** | **39.4** | **518.2** | **40.8** | **406.9** | **36.8** | **420.3** |
| Vis-50 | | 40.3 | **274.0** | 32.8 | **285.6** | 62.7 | **340.5** | 40.7 | **323.0** |
| | ✓ | **31.7** | 431.1 | **31.9** | 368.7 | **35.3** | 639.1 | **33.6** | 958.4 |
| Vis-100 | | 36.3 | 119.4 | 32.3 | **103.1** | 58.5 | 190.0 | 63.2 | **222.0** |
| | ✓ | **23.2** | **82.7** | **28.0** | 182.4 | **27.7** | **177.9** | **27.6** | 300.7 |
| Vis-200 | | 27.4 | 167.1 | 29.3 | **103.5** | 40.6 | **130.2** | 38.8 | **122.8** |
| | ✓ | **19.8** | **159.5** | **22.5** | 207.7 | **27.2** | 305.4 | **25.8** | 424.5 |

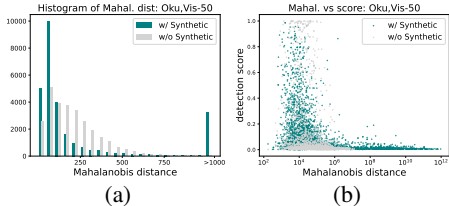

(a)                             (b)

Figure 3: **Change in distribution gap when synthetic images are used** for each image of the Okutama-Action dataset under the Vis-50 setup.

is small, the cross-domain presents similar or better accuracy than the same-domain. However, the effectiveness of the cross-domain continues to decrease as the number of real images increases.

These analyses strongly indicate that *synthetic data is effective in both same-domain and cross-domain tasks, particularly the impact of synthetic images is more significant when the amount of real data is small and then it continues to diminish as the amount of real data increases.* These findings fully confirm that synthetic data can greatly enhance learning in the data scarcity realm, where real data are hard to obtain, if adequately integrated into learning.

**Analysis in terms of distribution gaps.** Table 2 provides distribution gaps for the various test sets with and without synthetic data in a cross-domain setup. It is observed that for some cases, the use of synthetic data ('all' in the Table) unexpectedly increases the distribution gap compared to the cases without synthetic data. On the other hand, the distribution gap over half of the test images located closer to the reference dataset in terms of the Mahalanobis distance ('50%' in the Table) decreased as expected when synthetic data is added.

To investigate the change in the distribution gap in detail, we compare histograms representing the number of test images with respect to the Mahalanobis distance with and without synthetic data (Fig. 3a). Including synthetic data effectively reduces the Mahalanobis distance for most of the test images, yet the number of outliers with extremely large Mahalanobis distances also increases. We also compare how the distribution of test images with respect to detection scores *vs.* Mahalanobis distances differ depending on whether synthetic data is included or not (Fig. 3b). When using synthetic data in training, a majority of test images come with high detection confidence and small Mahalanobis distances contributing to a detection accuracy increase. However, the test images with large Mahalanobis distances and low detection confidence also appear more frequently. The analysis indicates that *a majority of synthetic data serves to improve the detector's ability for most test images in general with some exceptions of outlier images in a cross-domain setup.*

## 5.2 A Study on the Impact of Synthetic Data

For the second study, we explored the scalability behavior of synthetic data on the two impacts of using the synthetic data in training, mentioned in the first study. Specifically, we compare five cases with no synthetic images, 100, 500, 1000, and 2000 synthetic images in training in terms of the accuracy and distribution gap.

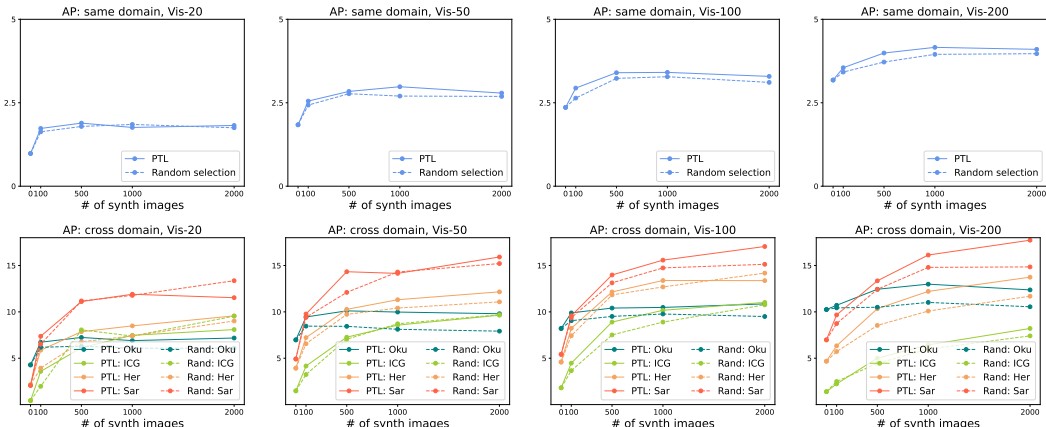

Figure 4: **Accuracy with the size of the synthetic dataset.** Plots in the top and bottom rows show APs in same-domain and cross-domain tasks, respectively.

**Analysis in terms of accuracy.** In this study, as PTL, our standard method of using synthetic images in training, gradually increases the number of synthetic images as training progresses, we investigate the scaling behavior of synthetic data by comparing models at different training checkpoints. To exclude the potential methodological influence of PTL in this general investigation, we also consider a random selection method, which randomly selects the same number of synthetic images as those used in PTL for training after applying the sim2real transformation.

In Fig. 4, two notable observations can be found regarding accuracy: i) in all setups, including same-domain and cross-domain, regardless of the method for synthetic data integration, the accuracy continues to increase while the rate of accuracy increase decreases as more synthetic images are used in training, and ii) as more real images are used in training, the checkpoint where the increase in accuracy rapidly diminishes usually occurs when a relatively large number of synthetic images are used. These observations indicate that *the impact of synthetic data continues to decrease as more synthetic images are included, but the capacity to use more synthetic data without sacrificing accuracy is expanded as more real data is used*.

**Analysis in terms of distribution gaps.** In Table 3 that shows the scalability behavior of synthetic data with respect to distribution gap, it is observed that the distribution gap mostly continues to decrease while the rate of change also decreases as more synthetic images are used in training. This is aligned to that of the previous analysis regarding accuracy.

In Fig 5, we can figure out how the distribution of test images for the detection score and Mahalanobis distance changed with the number of synthetic images used in training. Two notable observations are presented in the scatter plots: i) samples with high detection score ($>0.2$) appear more often as more synthetic images are used, and ii) samples with large Mahalanobis distance also appear more frequently when using a very large number of synthetic images (*i.e.*, 2000).

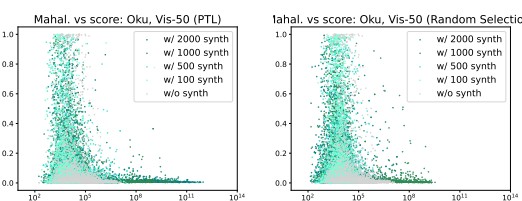

Figure 5: **Detection accuracy-distribution gap scatter plot with various numbers of synthetic images.** The left and right plots are made with PTL and random selection, respectively, using the Okutama-Action dataset under the Vis-50 setup. Darker dots represent test data when using more synthetic images for training.

**PTL *vs*. random selection.** In the previous analysis, two conflicting observations were found regarding the comparison between PTL and random selection. Firstly, PTL consistently provides better accuracy than random selection, regardless of training settings, for both the same domain and cross-domain tasks (Fig. 4). On the other hand, random selection is generally more effective in reducing the distribution gap using synthetic data than PTL (Tab. 3). When selecting synthetic images from the synthetic pool, PTL focuses more on synthetic images with similar characteristics to the reference dataset more frequently, rather than simply increasing the generalization ability of the training set. As this selection strategy is proposed to

Table 3: **Distribution gaps with various numbers of synthetic images.** 50% of the test images with the smallest Mahalanobis distance from the reference dataset are used for calculation.

(a) PTL

| dataset | Vis-20 | | | | | Vis-50 | | | | | Vis-100 | | | | | Vis-200 | | | | |
|---|---|---|---|---|---|---|---|---|---|---|---|---|---|---|---|---|---|---|---|---|
| | 0 | 100 | 500 | 1000 | 2000 | 0 | 100 | 500 | 1000 | 2000 | 0 | 100 | 500 | 1000 | 2000 | 0 | 100 | 500 | 1000 | 2000 |
| Okutama | 90.6 | 68.9 | 34.5 | 30.9 | 35.1 | 40.3 | 27.9 | 26.9 | 26.0 | 31.7 | 36.3 | 32.6 | 24.2 | 26.5 | 23.2 | 27.4 | 27.8 | 20.8 | 19.7 | 19.8 |
| ICG | 63.0 | 33.8 | 33.5 | 34.9 | 39.4 | 32.8 | 24.6 | 24.4 | 28.4 | 32.8 | 32.3 | 28.2 | 26.5 | 28.5 | 28.0 | 29.3 | 30.1 | 22.9 | 23.2 | 22.5 |
| HERIDAL | 151.0 | 85.7 | 40.4 | 38.0 | 40.8 | 62.7 | 82.2 | 32.5 | 31.9 | 35.3 | 58.5 | 43.7 | 34.5 | 39.1 | 27.7 | 40.6 | 46.9 | 26.8 | 31.9 | 27.2 |
| SARD | 132.2 | 101.9 | 36.5 | 37.0 | 36.8 | 40.7 | 32.8 | 31.4 | 30.0 | 33.6 | 63.2 | 41.0 | 32.6 | 46.4 | 27.6 | 38.8 | 40.4 | 27.8 | 32.5 | 25.8 |

(b) Random selection

| dataset | Vis-20 | | | | | Vis-50 | | | | | Vis-100 | | | | | Vis-200 | | | | |
|---|---|---|---|---|---|---|---|---|---|---|---|---|---|---|---|---|---|---|---|---|
| | 0 | 100 | 500 | 1000 | 2000 | 0 | 100 | 500 | 1000 | 2000 | 0 | 100 | 500 | 1000 | 2000 | 0 | 100 | 500 | 1000 | 2000 |
| Okutama | 90.6 | 31.4 | 26.1 | 29.1 | 27.2 | 40.3 | 31.1 | 21.9 | 19.7 | 22.0 | 36.3 | 24.3 | 20.4 | 20.0 | 18.4 | 27.4 | 25.8 | 19.4 | 16.0 | 15.5 |
| ICG | 63.0 | 106.0 | 25.8 | 29.8 | 28.8 | 32.8 | 53.2 | 23.0 | 21.4 | 23.1 | 32.3 | 24.4 | 23.9 | 24.7 | 22.4 | 29.3 | 26.7 | 22.2 | 20.0 | 20.3 |
| HERIDAL | 151.0 | 136.2 | 31.2 | 34.6 | 27.9 | 62.7 | 45.2 | 24.2 | 22.9 | 22.6 | 58.5 | 34.4 | 25.5 | 31.5 | 21.5 | 40.6 | 35.7 | 26.8 | 22.1 | 20.1 |
| SARD | 132.2 | 436.4 | 33.0 | 37.3 | 33.5 | 40.7 | 69.2 | 26.6 | 23.0 | 24.0 | 63.2 | 35.0 | 29.3 | 27.4 | 22.8 | 38.8 | 34.1 | 28.1 | 20.9 | 20.2 |

prevent degradation of the sim2real transformation quality, using higher-quality transformed synthetic images in training has a positive impact on increasing detection accuracy.

## 5.3 A STUDY ON THE IMPACT OF THE SYNTHETIC DATA POOL

For the third study, we explored the inherent properties of the synthetic data pool that in turn influence the use of synthetic data.

**Accuracy comparison *w.r.t.* rendering parameters.** Our synthetic data pool, the Archangel-Synthetic dataset, was built to show various human appearances captured with a virtual UAV by controlling several rendering parameters in a simulation space (altitudes and radii of camera location, camera's viewing angles, and human characters and poses). To examine the effect of each parameter on using synthetic data, we construct five subsets of the synthetic data pool, where each is built more sparsely for one parameter while fixing the values of other parameters. Each sub-pool includes the synthetic data with sparsely sampled altitudes (SAlt), radii (SRad), viewing angles (SAng), human characters (SCha), or human poses (SPos)[5].

In Table 4, we compare the detection accuracy of the original pool and its five subsets. 'SPos' exhibits significantly lower accuracy than the original, while the other four subsets show similar or even higher accuracy. In sampling the synthetic pool, reducing the variety of human poses significantly decreases detection accuracy as it leads to the inability to cover a wide range of human poses in test data. However, the decrease in accuracy is not observed when using subsets of synthetic data linked to the sparse sampling of other parameters.

**Properties of the synthetic pool.** We introduce several metrics to understand the variation in the ability of the synthetic pool to cover a variety of human appearances, depending on the rendering parameters, when used sparsely. We firstly consider *how densely data is located in the feature space* (density) and *how diverse the data distribution is in the feature space* (diversity). Specifically, the density and diversity of the pool $\mathcal{P}$ can be defined as below:

$$\bullet \text{ density: } \frac{1}{|\text{adj}(\mathcal{P})|} \sum_{\mathbf{p},\mathbf{q} \in \text{adj}(\mathcal{P})} f(\mathbf{p})^\top f(\mathbf{q}). \qquad \bullet \text{ diversity: } \frac{1}{|\mathcal{P}|} \sum_{\mathbf{x} \in \mathcal{P}} ||f(\mathbf{x}) - \mu||_2^k. \qquad (5, 6)$$

$f(\cdot)$ is the embedding in the feature space of the detector. Here, we use a detector trained without the synthetic data to avoid the influence of them used in training when measuring the properties of the pool. $\text{adj}(\mathcal{P})$ includes all data pairs associated with different neighboring values from each rendering parameter while others are fixed. Intuitively, a high $f(\mathbf{p})^\top f(\mathbf{q})$ in eq. 5 indicates that $\mathbf{p}$ and $\mathbf{q}$ lie close to each other in the feature space. $\mu$ is the mean feature over all data points in $\mathcal{P}$ (*i.e.*, $\mu = \sum_{\mathbf{x} \in \mathcal{P}} f(\mathbf{x})$). $k$ is a hyper-parameter that controls how each data point deviates from $\mu$ when calculating diversity. Higher $k$ leads to more weights on the data points away from $\mu$. (We use $k = 10$). We also consider the domain gap between the synthetic pool and the reference dataset, which can be calculated in a similar way to measuring the distribution gap (eq. 4).

In Table 4, 'SPos', which showed significantly lower accuracy than the original pool, has the following properties: higher density, less diversity, and closer domain gap to the reference dataset than the

---

[5]Details on how to build subsets of the synthetic data pool are provided in the supplementary material.

Table 4: **Comparison of various synthetic pools in terms of various aspects.** This comparison is performed with the Vis-20 setting.

| pool | # img | accuracy | | | | | property | | | distribution gap | | | |
|---|---|---|---|---|---|---|---|---|---|---|---|---|---|
| | | VisDrone | Okutama | ICG | HERIDAL | SARD | density | diversity | domain gap | Okutama | ICG | HERIDAL | SARD |
| original | 17,280 | 1.82 | 7.18 | 8.09 | 9.57 | 11.53 | 473.1 | 1.7e+15 | 135.8 | 35.1 | 39.4 | 40.8 | 36.8 |
| SAlt | 8,640 | 1.76 | 6.15 | 7.48 | 9.79 | 13.67 | 482.8 | 1.1e+15 | 149.3 | 32.9 | 37.9 | 38.9 | 35.0 |
| SRad | 8,640 | 1.84 | 7.56 | 6.58 | 9.21 | 12.97 | 481.5 | 1.6e+15 | 137.6 | 31.9 | 36.3 | 35.1 | 32.3 |
| SAng | 8,640 | 1.96 | 6.95 | 9.36 | 9.55 | 13.67 | 468.5 | 1.7e+15 | 141.3 | 33.0 | 37.4 | 36.2 | 35.7 |
| SCha | 8,640 | 1.84 | 7.89 | 6.85 | 10.02 | 13.11 | 481.9 | 1.5e+15 | 142.0 | 30.2 | 37.3 | 36.4 | 33.1 |
| SPos | 5,760 | **1.88** | **6.62** | **2.13** | **3.41** | **6.45** | **542.9** | **5.7e+14** | **124.0** | **35.5** | **40.3** | **51.1** | **49.1** |

original and other pools. This strongly indicates that *sampling synthetic data from a denser but less diverse distribution adversely affects using synthetic data in training, leading to low accuracy.* Moreover, *the small domain gap of the pool to the reference dataset does not have a positive effect on the cross-domain tasks* resulting in significantly low accuracy on ICG, HERIDAL, and SARD.

**Analysis in terms of distribution gaps.** In Table 4, using 'SPos' results in a larger distribution gap for cross-domain datasets than using other pools. This is aligned well with our previous analyses.

# 6 DISCUSSIONS

Through our comprehensive analysis based on the extensive experiments, we have brought to light valuable findings that have not been previously identified or have been used without an accurate understanding. Our findings are described as follows:

**1) General mechanism for acquiring domain generalization ability.** Our experiments show that in cross-domain tasks, synthetic data has a major impact on reducing the distribution gaps from reference dataset for most data, resulting in remarkable increases in cross-domain accuracy. On the other hand, a considerable number of outlier data points unexpectedly had very large distribution gaps. Outliers may arise due to i) insufficient diversity of reference data that serves as a standard for collecting synthetic data and training the sim2real transformer, or ii) the inherent limitation of synthetic data pool, which does not fully represent the entire cross-domain data. We further discuss the behavior of these factors (*i.e.,* reference (real) data, sim2real transformation, synthetic data pool) affecting the acquisition of domain generalization ability in the following findings.

**2) Relationship in scalability between synthetic data and real data.** Our experiments show that the more real images are used, the more positive is the impact of synthetic data on detection performance, not only in same-domain tasks but also in cross-domain tasks. Our experiments also indicate that as the amount of synthetic data used for training gradually increases, the accuracy continuously improves and then plateaus at some point. Impressively, the maximum number of synthetic images that can be used without accuracy plateauing increases as more real images are used. Therefore, to maximize the impact of synthetic data in training, it is important to increase the amount of not only synthetic data but also real data. Our findings on scalability may have some connections to the previous works (Richter et al., 2016; Ros et al., 2016; Lee et al., 2021) searching for an optimal ratio between real and synthetic data in a training batch.

**3) Sim2real transformation quality *vs*. distribution gap.** Which is more important: improving the sim2real transformation quality or reducing the distribution gap between datasets to acquire domain generalization ability? The answer to this question is that improving the sim2real transformation quality is more important. In our experiments comparing PTL and random selection, PTL designed to prevent the sim2real transformation quality degradation was less effective than random selection in reducing the distribution gaps. However, PTL consistently yields better accuracy than random selection in most experimental settings.

**4) Effect of synthetic data pool.** In our experiments, we analyzed how the properties of the synthetic data pool were related to the effectiveness of using synthetic data in training. Examining different properties of different synthetic data pools, we found that the density and diversity of pools are correlated with cross-domain detection accuracy. Therefore, we can select the optimal synthetic data pool to maximize the benefit of synthetic data by investigating the properties of the pool in advance.

In closing, we anticipate that our findings will lead to a significant increase in using synthetic data in training in an appropriate manner in future research.

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
