# A  PRELIMINARIES

## A.1  MODELING REPRESENTATION SPACE OF SIGMOID-BASED DETECTOR

In this section, we describe modeling the representation space of a sigmoid-based object detector by fitting a multivariate Gaussian distribution. We denote the random variable of the input and its label of a linear classifier as $\mathbf{x} \in \mathcal{X}$ and $y = \{y_c\}_{c=1,\cdots,C} \in \mathcal{Y}, y_c = \{0, 1\}$, respectively. Then, the posterior distribution defined by the linear classifier whose output is formed by the sigmoid function can be expressed as follows:

$$P(y_c = 1|\mathbf{x}) = \frac{1}{1 + \exp\left(w_c \mathbf{x} b_c\right)} = \frac{\exp\left(w_c \mathbf{x} + b_c\right)}{\exp\left(w_c \mathbf{x} + b_c\right) + 1}, \tag{1}$$

where $w_c$ and $b_c$ are the weights and bias of the linear classifier for a category $c$, respectively.

Gaussian Discriminant Analysis (GDA) models the posterior distribution of the classifier by assuming that the class conditional distribution ($P(\mathbf{x}|y)$) and the class prior distribution ($P(y)$) follow the multivariate Gaussian and the Bernoulli distributions, respectively, as follows:

$$P(\mathbf{x}|y_c = 0) = \mathcal{N}(\mu_0, \Sigma_0), \qquad P(\mathbf{x}|y_c = 1) = \mathcal{N}(\mu_1, \Sigma_1),$$
$$P(y_c = 0) = \beta_0 / \left(\beta_0 + \beta_1\right), \qquad P(y_c = 1) = \beta_1 / \left(\beta_0 + \beta_1\right), \tag{2}$$

where $\mu_{\{0,1\}}$ and $\Sigma_{\{0,1\}}$ are the mean and covariance of the multivariate Gaussian distribution, and $\beta_{\{0,1\}}$ is the unnormalized prior for the category $c$ and the background.

For the special case of GDA where all categories share the same covariance matrix (*i.e.*, $\Sigma_0 = \Sigma_1 = \Sigma_c$), known as Linear Discriminant Analysis (LDA), the posterior distribution ($P(y_c|\mathbf{x})$) can be expressed with $P(\mathbf{x}|y_c)$ and $P(y_c)$ as follows:

$$
\begin{aligned}
P(y_c = 1|\mathbf{x}) &= \frac{P(y_c = 1)P(\mathbf{x}|y_c = 1)}{P(y_c = 0)P(\mathbf{x}|y_c = 0) + P(y_c = 1)P(\mathbf{x}|y_c = 1)} \\
&= \frac{\exp\left((\mu_1 \mu_0)^\top \Sigma_c^1 \mathbf{x} - \frac{1}{2}\mu_1^\top \Sigma_c^1 \mu_1 + \frac{1}{2}\mu_0^\top \Sigma_c^1 \mu_0 + \ln \beta_1/\beta_0\right)}{\exp\left((\mu_1 \mu_0)^\top \Sigma_c^1 \mathbf{x} - \frac{1}{2}\mu_1^\top \Sigma_c^1 \mu_1 + \frac{1}{2}\mu_0^\top \Sigma_c^1 \mu_0 + \ln \beta_1/\beta_0\right) + 1}.
\end{aligned}
\tag{3}
$$

Note that the quadratic term is canceled out since the shared covariance matrix is used. The posterior distribution derived by GDA in eq. 3 then becomes equivalent to the posterior distribution of the linear classifier with the sigmoid function in eq. 1 when $w_c = (\mu_1 \mu_0)^\top \Sigma_c^1$ and $b_c = -\frac{1}{2}\mu_1^\top \Sigma_c^1 \mu_1 + \frac{1}{2}\mu_0^\top \Sigma_c^1 \mu_0 + \ln \beta_1/\beta_0$. This implies that the representation space formed by $\mathbf{x}$ can be modeled by a multivariate Gaussian distribution.

Based on the above derivation, if $\mathbf{x}$ is the output of the penultimate layer of an object detector for a region proposal, and a linear classifier defined by $w_c$ and $b_c$ is the last layer of the object detector, it can be said that the representation space of the object detector for a category $c$ can be modeled with a multivariate Gaussian distribution. In other words, the representation space for a category $c$ can be represented by two parameters $\mu_1$ (*i.e.*, $\mu_c$) and $\Sigma_c$ of the multivariate Gaussian distribution.

**Discussion.** The sigmoid function can be viewed as a special case of the softmax function defined for a single category as both functions take the form of an exponential term for the category-of-interest normalized by the sum of exponential terms for all considered categories. Therefore, it is straightforward to derive the modeling for the sigmoid-based detector from the previous work Lee et al. (2018), who shows that the softmax-based classifier can be modeled with a multivariate Gaussian distribution in the representation space. However, our derivation is still meaningful in that it extends the applicability of an existing modeling limited to a certain type of classifier (*i.e.*, based on softmax) to general object detectors (*i.e.*, based on sigmoid). Most object detectors, especially one-stage detectors, generally use the sigmoid function, which does not consider other categories when calculating the model output for a certain category, since more than one category can be active on a single output.

## A.2 CROSS-ENTROPY WITH MIXTURE OF DELTA DISTRIBUTIONS AND MULTIVARIATE GAUSSIAN DISTRIBUTION

In this section, we derive the cross-entropy of two distributions that are modeled by a mixture of delta distributions and a multivariate Gaussian distribution as the normalized sum of Mahalanobis distances. Assume that the data distributions $P$ and $Q$ in two sets $\mathcal{D}_P$ and $\mathcal{D}_Q$ can be modeled by density functions ($p$ and $q$) that take the form of a mixture of delta distributions and a multivariate Gaussian distribution, respectively, as follows:

$$p(\mathbf{x}) = \frac{1}{|\mathcal{D}_P|} \sum_{\mathbf{x}' \in \mathcal{D}_P} \delta(\mathbf{x} - \mathbf{x}'), \tag{4}$$

$$q(\mathbf{x}) = \frac{1}{\sqrt{2\pi}^k \det(\Sigma)^{1/2}} \exp\left(-\frac{1}{2}(\mathbf{x} - \mu)^\top \Sigma^{-1}(\mathbf{x} - \mu)\right), \tag{5}$$

where $\delta(\mathbf{x})$ is a Dirac delta function whose value is zero everywhere except at $\mathbf{x} = \mathbf{0}$ and whose integral over $\mathcal{X}$, which is the entire space of $\mathbf{x}$, is one. $\mu$ and $\Sigma$ are two parameters of the multivariate Gaussian distribution, which can be calculated empirically over all $\mathbf{x} \in \mathcal{D}_Q$.

Then, the cross entropy, which statistically measures the difference from $Q$ to $P$ where $Q$ is treated as the reference distribution, can be expressed as:

$$\mathcal{H}(P,Q) = -\int_{\mathcal{X}} p(\mathbf{x}) \ln q(\mathbf{x}) d\mathbf{x}$$

$$= -\int_{\mathcal{X}} \frac{1}{|\mathcal{D}_P|} \sum_{\mathbf{x}' \in \mathcal{D}_P} \delta(\mathbf{x} - \mathbf{x}') \ln\left(\frac{1}{\sqrt{2\pi}^k \det(\Sigma)^{1/2}} \exp\left(-\frac{1}{2}(\mathbf{x} - \mu)^\top \Sigma^{-1}(\mathbf{x} - \mu)\right)\right) d\mathbf{x}. \tag{6}$$

Using two basic rules of i) $\int_{\mathcal{X}} \left(\sum_n f_n(x)\right) dx = \sum_n \left(\int_{\mathcal{X}} f_n(x) dx\right)$ if the summation is performed on a finite set, and ii) $\int_{\mathcal{X}} \delta(x - a) f(x) dx = f(a)$ if $f(x)$ is continuous on $\mathcal{X}$, the cross entropy in eq. 6 can be derived as:

$$\mathcal{H}(P,Q) = -\frac{1}{|\mathcal{D}_p|} \sum_{\mathbf{x} \in \mathcal{D}_p} \ln\left(\frac{1}{\sqrt{2\pi}^k \det(\Sigma)^{1/2}} \exp\left(-\frac{1}{2}(\mathbf{x} - \mu)^\top \Sigma^{-1}(\mathbf{x} - \mu)\right)\right)$$

$$= \frac{1}{2|\mathcal{D}_p|} \sum_{\mathbf{x} \in \mathcal{D}_p} (\mathbf{x} - \mu)^\top \Sigma^{-1}(\mathbf{x} - \mu) + \ln\left(\sqrt{2\pi}^k \det(\Sigma)^{1/2}\right). \tag{7}$$

Note that conditions for realizing the two basic rules are satisfied in our scenario, as i) the summation is computed on a finite set $\mathcal{D}_P$, and ii) a log of the multivariate Gaussian distribution is continuous on $\mathcal{X}$.

In our scenario where the cross-entropy is used to compare the distribution gaps of different test datasets (here, $\mathcal{D}_P$s) while the reference dataset ($\mathcal{D}_Q$) is fixed, and it is computed on the representation space of the detector, the cross-entropy can be expressed as:

$$\mathcal{H}(P,Q) = \frac{1}{2|\mathcal{D}_P|} \sum_{\mathbf{x} \in \mathcal{D}_P} (f(\mathbf{x}) - \mu)^\top \Sigma^{-1}(f(\mathbf{x}) - \mu) + C, \tag{8}$$

where $f(\cdot)$ is the output of the detector in the representation space. The second term of eq. 7 can be regarded as a constant since $k$ (the dimension of the representation space) and $\Sigma$ (parameter of the reference dataset $\mathcal{D}_Q$) are not affected by the test dataset $\mathcal{D}_P$.

## B IMPLEMENTATION DETAILS

**PTL.** We followed the original PTL paper Shen et al. (2023) for all architectural details and training specifications of PTL except for the numbers of training epochs and iterations. The numbers of training epochs (used in sim2real transformer training) and training iterations (used in detector training) are modified to adopt a training time curtailment strategy. Specifically, in the original PTL,

Table 1: ***Wall-clock* Training time breakdown** for sim2real transformer training and detector training. Training time is shown in *mins*. The numbers in the parentheses indicate training epochs and iterations for the corresponding PTL iteration for sim2real transformer training and detector training, respectively.

(a) sim2real transformer, Vis-20

| from-prev-iter | 0 | 1 | 2 | 3 | 4 |
|---|---|---|---|---|---|
|  | 28 (100) | 41 (100) | 56 (100) | 69 (100) | 83 (100) |
| ✓ | 28 (100) | 8 (20) | 12 (20) | 14 (20) | 17 (20) |

(b) detector, Vis-20

| from-prev-iter | 0 | 1 | 2 | 3 | 4 | 5 |
|---|---|---|---|---|---|---|
|  | 40 (6.0k) | 36 (6.0k) | 32 (6.0k) | 28 (6.0k) | 25 (6.0k) | 22 (6.0k) |
| ✓ | 40 (6.0k) | 24 (1.2k) | 21 (1.2k) | 19 (1.2k) | 17 (1.2k) | 15 (1.2k) |

(c) sim2real transformer, Vis-50

| from-prev-iter | 0 | 1 | 2 | 3 | 4 |
|---|---|---|---|---|---|
|  | 87 (100) | 101 (100) | 114 (100) | 128 (100) | 142 (100) |
| ✓ | 87 (100) | 20 (20) | 23 (20) | 26 (20) | 29 (20) |

(d) detector, Vis-50

| from-prev-iter | 0 | 1 | 2 | 3 | 4 | 5 |
|---|---|---|---|---|---|---|
|  | 40 (6.0k) | 38 (6.0k) | 36 (6.0k) | 35 (6.0k) | 33 (6.0k) | 31 (6.0k) |
| ✓ | 40 (6.0k) | 25 (1.2k) | 24 (1.2k) | 23 (1.2k) | 22 (1.2k) | 21 (1.2k) |

the sim2real transformer and detector are trained for 100 epochs and 6.0k iterations, respectively, but when adopting the strategy, they are trained for 20 epochs and 1.2k iterations, respectively, after the 0th iteration.

**Random selection.** For random selection, we used the PTL implementation after modifying the synthetic data selection. In particular, while PTL is designed to select synthetic images by weighting images closer in domain gap to the training set, this selection is modified to randomly select synthetic data. All other parts except this selection process of the PTL training pipeline were used unchanged.

**Subsets of synthetic data pool** The Archangel-synthetic dataset **?** was originally created by varying the five rendering parameters as follows: 10 altitudes (from $5m$ to $50m$ at $5m$ interval), 6 radii (from $5m$ to $30m$ at $5m$ interval), 12 angles (from $0°$ to $330°$ at $30°$ interval) 8 human characters (Juliet, Kelly, Lucy, Mary, Romeo, Scott, Troy, and Victor), and 3 human poses (stand, prone, squat). Each subset of the synthetic data pool is built using a sparse set of each rendering parameter, as follows:

- SAlt: using sparser 5 altitudes from $10m$ to $50m$ at $10m$ interval.
- SRad: using sparser 3 radii from $10m$ to $30m$ at $10m$ interval.
- SAng: using sparser 6 angles from $0°$ to $300°$ at $60°$ interval.
- SCha: using sparser 4 human characters of Juliet, Kelly, Romeo, and Scott.
- SPos: using sparser 1 human pose of standing.

For each subset, all other parameters were the same as those of the original pool, except for the rendering parameter indicated to be used sparsely.

## C  NUMERICAL RESULTS

In this section, we present the numerical results of the graphs used for analysis in the main manuscript and additional results not presented in the main manuscript.

### C.1  CURTAILMENT OF PTL TRAINING TIME

The reduced training time and altered accuracy by adopting the *tune-from-previous-iteration* strategy is reported in the main manuscript. Here, we also present the reduced time for two separate components of the PTL training pipeline that are affected by the strategy: detector training and sim2real transformer training (Table 1). The corresponding training times (in *mins*) with and without the strategy for every PTL iteration in Vis-20/50 settings are shown in the table. It is noteworthy that training time per PTL iteration is longer in Vis-50 than Vis-20 when using the same numbers of training iterations and epochs. This is because in our experimental setting, the real images are larger in size than the synthetic images (the image sizes of VisDrone images used as real data and the Archangel-Synthetic images used as synthetic data are 2000×1500 and 512×512, respectively) and thus require more computation time, and account for a larger portion of the training set in Vis-50.

Table 2: **Numerical results with the size of real dataset**. In each bin presenting accuracy, the mean and standard deviation of AP@.5 and AP@[.5:.95] calculated over 3 runs are reported.

(a) Same-domain and cross-domain accuracy in the Vis-$N$

| setup | w/ synth | VisDrone | Okutama | ICG | HERIDAL | SARD |
|---|---|---|---|---|---|---|
| Vis-20 | | $3.43_{\pm0.57}$ / $0.98_{\pm0.12}$ | $18.38_{\pm8.74}$ / $4.73_{\pm2.61}$ | $2.14_{\pm0.70}$ / $0.43_{\pm0.15}$ | $7.11_{\pm3.45}$ / $2.13_{\pm1.08}$ | $7.24_{\pm3.32}$ / $2.07_{\pm1.14}$ |
| | ✓ | $6.18_{\pm0.47}$ / $1.82_{\pm0.33}$ | $29.93_{\pm3.01}$ / $7.18_{\pm0.90}$ | $29.30_{\pm2.70}$ / $8.09_{\pm0.33}$ | $28.61_{\pm2.91}$ / $9.57_{\pm1.13}$ | $34.96_{\pm3.26}$ / $11.53_{\pm1.06}$ |
| Vis-50 | | $6.13_{\pm0.28}$ / $1.84_{\pm0.21}$ | $25.65_{\pm4.62}$ / $6.98_{\pm1.56}$ | $6.57_{\pm2.41}$ / $1.48_{\pm0.89}$ | $12.12_{\pm3.35}$ / $3.93_{\pm1.17}$ | $17.73_{\pm1.58}$ / $4.92_{\pm0.61}$ |
| | ✓ | $8.91_{\pm0.20}$ / $2.79_{\pm0.08}$ | $37.67_{\pm0.59}$ / $9.80_{\pm0.25}$ | $32.86_{\pm5.36}$ / $9.64_{\pm2.03}$ | $36.88_{\pm3.79}$ / $12.16_{\pm1.93}$ | $45.75_{\pm2.16}$ / $15.92_{\pm1.73}$ |
| Vis-100 | | $7.91_{\pm0.13}$ / $2.36_{\pm0.07}$ | $31.37_{\pm1.49}$ / $8.21_{\pm0.29}$ | $7.60_{\pm1.51}$ / $1.81_{\pm0.32}$ | $14.64_{\pm6.04}$ / $4.59_{\pm1.47}$ | $18.27_{\pm1.25}$ / $5.42_{\pm0.50}$ |
| | ✓ | $10.56_{\pm0.49}$ / $3.29_{\pm0.23}$ | $41.18_{\pm3.35}$ / $10.86_{\pm1.07}$ | $35.69_{\pm1.65}$ / $11.02_{\pm1.35}$ | $38.54_{\pm7.75}$ / $13.37_{\pm2.36}$ | $48.15_{\pm1.18}$ / $17.05_{\pm0.87}$ |
| Vis-200 | | $10.55_{\pm1.41}$ / $3.18_{\pm0.55}$ | $38.58_{\pm4.81}$ / $10.25_{\pm1.66}$ | $6.50_{\pm3.17}$ / $1.39_{\pm0.63}$ | $14.76_{\pm9.76}$ / $4.68_{\pm2.72}$ | $21.87_{\pm7.48}$ / $6.98_{\pm1.97}$ |
| | ✓ | $12.78_{\pm0.48}$ / $4.10_{\pm0.28}$ | $46.62_{\pm0.93}$ / $12.37_{\pm0.21}$ | $30.48_{\pm0.34}$ / $8.21_{\pm0.43}$ | $37.60_{\pm2.12}$ / $13.74_{\pm1.46}$ | $49.60_{\pm1.78}$ / $17.73_{\pm0.49}$ |

(b) Same-domain accuracy w/o synthetic data and its ratio to the Vis-$N$ (w/ synthetic data) when using the same number of real images

| | | # of real image | | | |
|---|---|---|---|---|---|
| | testset | 20 | 50 | 100 | 200 |
| accuracy | Okutama | $37.93_{\pm1.75}$ / $9.93_{\pm0.16}$ | $51.31_{\pm2.05}$ / $14.13_{\pm0.90}$ | $55.98_{\pm0.75}$ / $16.68_{\pm0.45}$ | $64.76_{\pm0.97}$ / $20.28_{\pm0.17}$ |
| ratio to Vis-$N$ (w/ synth) | | 0.79 / 0.72 | 0.73 / 0.69 | 0.74 / 0.65 | 0.72 / 0.61 |
| accuracy | ICG | $40.36_{\pm0.96}$ / $10.63_{\pm0.38}$ | $60.95_{\pm1.76}$ / $19.57_{\pm1.19}$ | $73.23_{\pm1.50}$ / $27.53_{\pm0.95}$ | $84.21_{\pm1.50}$ / $35.98_{\pm1.20}$ |
| ratio to Vis-$N$ (w/ synth) | | 0.73 / 0.76 | 0.54 / 0.49 | 0.49 / 0.40 | 0.36 / 0.23 |
| accuracy | HERIDAL | $41.39_{\pm2.86}$ / $12.75_{\pm1.82}$ | $58.97_{\pm2.86}$ / $19.76_{\pm0.96}$ | $65.78_{\pm0.70}$ / $26.27_{\pm1.25}$ | $71.53_{\pm0.49}$ / $31.18_{\pm1.70}$ |
| ratio to Vis-$N$ (w/ synth) | | 0.69 / 0.75 | 0.63 / 0.62 | 0.59 / 0.51 | 0.53 / 0.44 |
| accuracy | SARD | $33.44_{\pm6.36}$ / $8.52_{\pm2.20}$ | $51.35_{\pm1.68}$ / $15.28_{\pm0.78}$ | $66.81_{\pm3.15}$ / $23.75_{\pm1.79}$ | $75.76_{\pm1.62}$ / $30.17_{\pm0.98}$ |
| ratio to Vis-$N$ (w/ synth) | | 1.05 / 1.35 | 0.89 / 1.04 | 0.72 / 0.72 | 0.65 / 0.59 |

## C.2 SCALABILITY BEHAVIOR OF REAL DATA

In Table 2, we present numerical results used to generate Fig. 2 of the main manuscript. Specifically, the numbers in Table 2a correspond to Fig. 2(a) and (b) of the main manuscript while the numbers in Table 2b are matched to Fig. 2(c). The results using AP@.5 follow a similar trend to those using AP@[.5:.95], which have been analyzed in the main manuscript.

## C.3 SCALABILITY BEHAVIOR OF SYNTHETIC DATA

In Table 3, we present numerical results used to generate Fig. 4 of the main manuscript. The results using AP@.5 follows a similar trend to those using AP@[.5:.95], which have been analyzed in the main manuscript.

In Figure 1, we also show the scatter plots of detection scores and Mahalanobis distances for different numbers of synthetic images used in training. Among different experimental settings, we present the scatter plots for three cases: i) testing on the Okutama-Action dataset in Vis-50, ii) testing on the SARD dataset in Vis-100, and iii) testing on the HERIDAL dataset in Vis-200. For reference, the first case is the same as the scatter plots in Fig. 3(b) and Fig. 5 of the main manuscript. To better focus on the distribution of each scatter plot, scatter plots are shown separately for each size of synthetic data. In the main manuscript, these scatterplots are shown together in one figure to emphasize the differences between the plots. The observations of change in the scatter plots for the other two cases are similar to those in the first case, which has been analyzed in the main manuscript.

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

Table 3: **Numerical results with the size of synthetic dataset.** 'Random' denotes random selection.

(a) Vis-20

| method | test set | 0 | 100 | 500 | 1000 | 2000 |
|---|---|---|---|---|---|---|
| | | | | # of synthetic image | | |
| PTL | VisDrone | $3.43_{\pm0.57}$ / $0.98_{\pm0.12}$ | $5.88_{\pm0.58}$ / $1.73_{\pm0.21}$ | $6.48_{\pm0.56}$ / $1.89_{\pm0.13}$ | $6.28_{\pm0.86}$ / $1.76_{\pm0.28}$ | $6.18_{\pm0.47}$ / $1.82_{\pm0.33}$ |
| Random | | | $5.90_{\pm0.29}$ / $1.63_{\pm0.07}$ | $6.40_{\pm0.85}$ / $1.79_{\pm0.28}$ | $6.41_{\pm0.20}$ / $1.85_{\pm0.09}$ | $6.09_{\pm0.69}$ / $1.75_{\pm0.28}$ |
| PTL | Okutama | $18.38_{\pm8.74}$ / $4.73_{\pm2.61}$ | $28.01_{\pm2.89}$ / $6.73_{\pm0.80}$ | $30.54_{\pm1.06}$ / $7.24_{\pm0.36}$ | $29.63_{\pm1.21}$ / $6.90_{\pm0.41}$ | $29.93_{\pm3.01}$ / $7.18_{\pm0.90}$ |
| Random | | | $27.54_{\pm1.40}$ / $6.18_{\pm0.62}$ | $28.20_{\pm1.74}$ / $6.30_{\pm0.34}$ | $27.20_{\pm0.90}$ / $6.08_{\pm0.24}$ | $27.80_{\pm3.03}$ / $6.08_{\pm0.99}$ |
| PTL | ICG | $2.14_{\pm0.70}$ / $0.43_{\pm0.15}$ | $15.16_{\pm3.85}$ / $3.58_{\pm1.24}$ | $23.03_{\pm4.68}$ / $6.13_{\pm1.79}$ | $26.70_{\pm1.86}$ / $7.47_{\pm1.37}$ | $29.30_{\pm2.70}$ / $8.09_{\pm0.33}$ |
| Random | | | $8.97_{\pm0.67}$ / $1.97_{\pm0.20}$ | $29.62_{\pm2.03}$ / $8.06_{\pm0.30}$ | $26.69_{\pm6.21}$ / $7.39_{\pm1.14}$ | $33.70_{\pm0.85}$ / $9.54_{\pm0.87}$ |
| PTL | HERIDAL | $7.11_{\pm3.45}$ / $2.13_{\pm1.08}$ | $20.24_{\pm3.85}$ / $5.87_{\pm1.83}$ | $26.50_{\pm3.02}$ / $7.86_{\pm1.10}$ | $26.98_{\pm5.38}$ / $8.49_{\pm1.14}$ | $28.61_{\pm2.91}$ / $9.57_{\pm1.13}$ |
| Random | | | $14.82_{\pm2.57}$ / $3.94_{\pm1.00}$ | $23.37_{\pm2.19}$ / $6.77_{\pm0.92}$ | $25.98_{\pm2.92}$ / $7.41_{\pm0.73}$ | $29.22_{\pm4.55}$ / $9.02_{\pm1.95}$ |
| PTL | SARD | $7.24_{\pm3.32}$ / $2.07_{\pm1.14}$ | $24.51_{\pm3.39}$ / $7.37_{\pm1.33}$ | $34.74_{\pm1.93}$ / $11.11_{\pm1.14}$ | $35.65_{\pm3.07}$ / $11.90_{\pm0.78}$ | $34.96_{\pm3.26}$ / $11.53_{\pm1.06}$ |
| Random | | | $22.15_{\pm3.13}$ / $6.52_{\pm1.44}$ | $34.98_{\pm5.04}$ / $11.18_{\pm1.77}$ | $37.35_{\pm3.35}$ / $11.78_{\pm0.67}$ | $40.01_{\pm2.07}$ / $13.36_{\pm0.85}$ |

(b) Vis-50

| method | test set | 0 | 100 | 500 | 1000 | 2000 |
|---|---|---|---|---|---|---|
| | | | | # of synthetic image | | |
| PTL | VisDrone | $6.13_{\pm0.28}$ / $1.84_{\pm0.21}$ | $8.48_{\pm0.26}$ / $2.55_{\pm0.10}$ | $9.27_{\pm0.29}$ / $2.84_{\pm0.12}$ | $9.39_{\pm0.12}$ / $2.98_{\pm0.07}$ | $8.91_{\pm0.20}$ / $2.79_{\pm0.08}$ |
| Random | | | $8.17_{\pm0.28}$ / $2.43_{\pm0.22}$ | $9.23_{\pm0.29}$ / $2.77_{\pm0.15}$ | $8.97_{\pm0.08}$ / $2.70_{\pm0.11}$ | $9.01_{\pm0.56}$ / $2.69_{\pm0.04}$ |
| PTL | Okutama | $25.65_{\pm4.62}$ / $6.98_{\pm1.56}$ | $35.21_{\pm4.67}$ / $9.45_{\pm1.70}$ | $37.94_{\pm1.84}$ / $9.88_{\pm0.76}$ | $37.17_{\pm2.10}$ / $9.63_{\pm0.95}$ | $38.85_{\pm3.34}$ / $10.04_{\pm1.10}$ |
| Random | | | $32.66_{\pm5.86}$ / $8.46_{\pm1.92}$ | $34.48_{\pm5.68}$ / $8.44_{\pm2.03}$ | $33.68_{\pm6.44}$ / $8.12_{\pm2.04}$ | $33.29_{\pm4.53}$ / $7.92_{\pm1.20}$ |
| PTL | ICG | $6.57_{\pm2.41}$ / $1.48_{\pm0.89}$ | $16.87_{\pm2.23}$ / $4.16_{\pm0.71}$ | $29.70_{\pm3.13}$ / $7.27_{\pm1.27}$ | $30.94_{\pm6.70}$ / $8.57_{\pm2.54}$ | $32.86_{\pm5.36}$ / $9.64_{\pm2.03}$ |
| Random | | | $14.43_{\pm2.72}$ / $3.24_{\pm0.90}$ | $28.78_{\pm4.90}$ / $7.05_{\pm1.65}$ | $31.45_{\pm1.49}$ / $8.72_{\pm1.44}$ | $35.51_{\pm2.12}$ / $9.69_{\pm0.33}$ |
| PTL | HERIDAL | $12.12_{\pm3.35}$ / $3.93_{\pm1.17}$ | $22.81_{\pm1.43}$ / $7.22_{\pm0.59}$ | $31.62_{\pm1.32}$ / $10.27_{\pm0.74}$ | $33.24_{\pm1.66}$ / $11.31_{\pm1.30}$ | $36.88_{\pm3.79}$ / $12.16_{\pm1.93}$ |
| Random | | | $21.71_{\pm2.14}$ / $6.56_{\pm0.72}$ | $29.87_{\pm3.93}$ / $9.73_{\pm1.39}$ | $32.11_{\pm5.48}$ / $10.41_{\pm2.96}$ | $34.06_{\pm5.59}$ / $11.08_{\pm2.31}$ |
| PTL | SARD | $17.73_{\pm1.58}$ / $4.92_{\pm0.61}$ | $32.18_{\pm3.75}$ / $9.77_{\pm1.06}$ | $43.67_{\pm4.01}$ / $14.33_{\pm1.66}$ | $43.59_{\pm2.31}$ / $14.15_{\pm2.17}$ | $45.75_{\pm2.16}$ / $15.92_{\pm1.73}$ |
| Random | | | $30.74_{\pm1.61}$ / $9.41_{\pm0.65}$ | $38.52_{\pm0.88}$ / $12.10_{\pm0.96}$ | $44.28_{\pm2.83}$ / $14.30_{\pm1.68}$ | $45.56_{\pm3.26}$ / $15.21_{\pm1.64}$ |

(c) Vis-100

| method | test set | 0 | 100 | 500 | 1000 | 2000 |
|---|---|---|---|---|---|---|
| | | | | # of synthetic image | | |
| PTL | VisDrone | $7.91_{\pm0.13}$ / $2.36_{\pm0.07}$ | $9.58_{\pm0.57}$ / $2.94_{\pm0.21}$ | $10.79_{\pm0.28}$ / $3.40_{\pm0.09}$ | $10.82_{\pm0.45}$ / $3.41_{\pm0.07}$ | $10.56_{\pm0.49}$ / $3.29_{\pm0.23}$ |
| Random | | | $9.13_{\pm0.60}$ / $2.64_{\pm0.16}$ | $10.66_{\pm0.10}$ / $3.23_{\pm0.09}$ | $10.67_{\pm0.43}$ / $3.28_{\pm0.14}$ | $10.13_{\pm0.27}$ / $3.11_{\pm0.03}$ |
| PTL | Okutama | $31.37_{\pm1.49}$ / $8.21_{\pm0.29}$ | $36.61_{\pm4.46}$ / $9.89_{\pm1.28}$ | $40.37_{\pm4.12}$ / $10.41_{\pm0.91}$ | $40.76_{\pm4.69}$ / $10.48_{\pm1.12}$ | $41.18_{\pm3.35}$ / $10.86_{\pm1.07}$ |
| Random | | | $34.92_{\pm2.86}$ / $9.05_{\pm0.85}$ | $38.12_{\pm2.66}$ / $9.51_{\pm0.42}$ | $38.80_{\pm2.42}$ / $9.77_{\pm0.26}$ | $38.18_{\pm1.79}$ / $9.50_{\pm0.40}$ |
| PTL | ICG | $7.60_{\pm1.51}$ / $1.81_{\pm0.32}$ | $18.19_{\pm3.47}$ / $4.47_{\pm0.92}$ | $31.81_{\pm3.63}$ / $8.90_{\pm1.40}$ | $35.38_{\pm8.43}$ / $10.17_{\pm2.30}$ | $35.69_{\pm1.65}$ / $11.02_{\pm1.35}$ |
| Random | | | $16.33_{\pm1.34}$ / $3.66_{\pm0.57}$ | $31.67_{\pm3.23}$ / $7.51_{\pm1.20}$ | $32.98_{\pm2.74}$ / $8.90_{\pm1.23}$ | $38.75_{\pm2.49}$ / $10.76_{\pm0.97}$ |
| PTL | HERIDAL | $14.64_{\pm6.04}$ / $4.59_{\pm1.47}$ | $25.14_{\pm9.32}$ / $8.23_{\pm2.54}$ | $35.00_{\pm6.30}$ / $12.15_{\pm1.58}$ | $37.31_{\pm4.15}$ / $13.38_{\pm0.73}$ | $38.54_{\pm7.75}$ / $13.37_{\pm2.36}$ |
| Random | | | $23.77_{\pm7.78}$ / $7.43_{\pm2.13}$ | $34.28_{\pm7.17}$ / $11.83_{\pm2.37}$ | $35.89_{\pm4.74}$ / $12.69_{\pm1.33}$ | $40.59_{\pm5.17}$ / $14.18_{\pm2.50}$ |
| PTL | SARD | $18.27_{\pm1.25}$ / $5.42_{\pm0.50}$ | $31.16_{\pm5.12}$ / $9.58_{\pm1.86}$ | $42.93_{\pm2.58}$ / $13.98_{\pm1.20}$ | $46.04_{\pm2.30}$ / $15.58_{\pm0.73}$ | $48.15_{\pm1.18}$ / $17.05_{\pm0.87}$ |
| Random | | | $30.55_{\pm2.77}$ / $9.37_{\pm0.68}$ | $40.69_{\pm0.34}$ / $13.13_{\pm0.12}$ | $44.10_{\pm3.84}$ / $14.74_{\pm1.62}$ | $45.62_{\pm5.05}$ / $15.13_{\pm2.15}$ |

(d) Vis-200

| method | test set | 0 | 100 | 500 | 1000 | 2000 |
|---|---|---|---|---|---|---|
| | | | | # of synthetic image | | |
| PTL | VisDrone | $10.55_{\pm1.41}$ / $3.18_{\pm0.55}$ | $11.65_{\pm0.87}$ / $3.55_{\pm0.35}$ | $12.74_{\pm0.90}$ / $3.99_{\pm0.35}$ | $12.96_{\pm0.68}$ / $4.16_{\pm0.43}$ | $12.78_{\pm0.48}$ / $4.10_{\pm0.28}$ |
| Random | | | $11.35_{\pm1.23}$ / $3.42_{\pm0.52}$ | $12.07_{\pm1.11}$ / $3.72_{\pm0.46}$ | $12.63_{\pm0.59}$ / $3.95_{\pm0.36}$ | $12.59_{\pm1.17}$ / $3.97_{\pm0.49}$ |
| PTL | Okutama | $38.58_{\pm4.81}$ / $10.25_{\pm1.66}$ | $40.23_{\pm2.20}$ / $10.71_{\pm1.12}$ | $45.56_{\pm0.44}$ / $12.44_{\pm0.51}$ | $47.79_{\pm2.47}$ / $12.99_{\pm0.39}$ | $46.62_{\pm0.93}$ / $12.37_{\pm0.21}$ |
| Random | | | $39.99_{\pm2.82}$ / $10.43_{\pm0.48}$ | $39.95_{\pm2.33}$ / $10.50_{\pm0.71}$ | $41.88_{\pm1.66}$ / $11.02_{\pm0.74}$ | $41.20_{\pm0.92}$ / $10.55_{\pm0.03}$ |
| PTL | ICG | $6.50_{\pm3.17}$ / $1.39_{\pm0.63}$ | $9.50_{\pm1.65}$ / $2.22_{\pm0.44}$ | $20.56_{\pm2.32}$ / $5.01_{\pm0.40}$ | $25.67_{\pm4.82}$ / $6.44_{\pm1.65}$ | $30.48_{\pm0.34}$ / $8.21_{\pm0.43}$ |
| Random | | | $9.27_{\pm2.16}$ / $2.49_{\pm0.16}$ | $18.29_{\pm5.19}$ / $4.45_{\pm1.31}$ | $23.29_{\pm5.46}$ / $5.98_{\pm1.31}$ | $27.29_{\pm4.52}$ / $7.41_{\pm1.46}$ |
| PTL | HERIDAL | $14.76_{\pm9.76}$ / $4.68_{\pm2.72}$ | $19.87_{\pm3.79}$ / $6.34_{\pm0.69}$ | $30.51_{\pm6.31}$ / $10.34_{\pm1.39}$ | $34.76_{\pm8.89}$ / $12.21_{\pm2.76}$ | $37.60_{\pm2.12}$ / $13.74_{\pm1.46}$ |
| Random | | | $17.66_{\pm7.26}$ / $5.72_{\pm2.29}$ | $26.26_{\pm9.95}$ / $8.55_{\pm3.30}$ | $29.62_{\pm6.13}$ / $10.09_{\pm1.44}$ | $33.74_{\pm8.06}$ / $11.69_{\pm2.56}$ |
| PTL | SARD | $21.87_{\pm7.48}$ / $6.98_{\pm1.97}$ | $30.17_{\pm4.28}$ / $9.67_{\pm1.03}$ | $40.01_{\pm2.13}$ / $13.35_{\pm0.57}$ | $46.91_{\pm5.03}$ / $16.13_{\pm1.69}$ | $49.60_{\pm1.35}$ / $17.73_{\pm0.49}$ |
| Random | | | $27.51_{\pm5.72}$ / $8.74_{\pm1.69}$ | $38.96_{\pm5.45}$ / $12.43_{\pm2.31}$ | $44.59_{\pm1.56}$ / $14.80_{\pm0.82}$ | $43.25_{\pm5.80}$ / $14.85_{\pm2.10}$ |

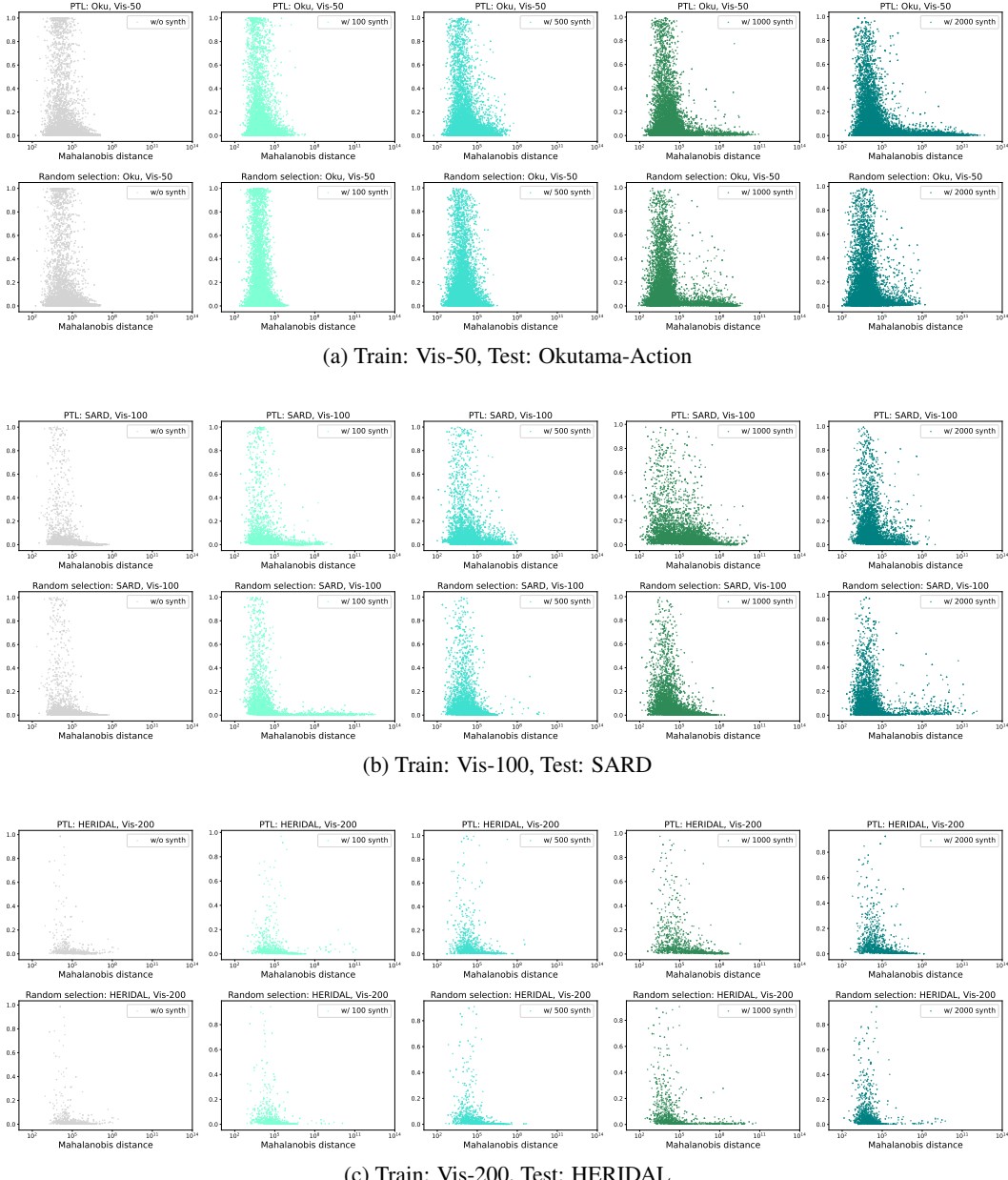

Figure 1: **Scatter plot of detection scores and Mahalanobis distances** with various numbers of synthetic images. For each case, plots in the first row and the second row represent the scatter results for PTL and random selection, respectively. Each of the five plots in each row shows the results without using synthetic images, or with using 100, 500, 1000, or 2000 synthetic images, in order.