# OpenReview forum: "Delving Deep into Sim2Real Transformation: Maximizing Impact of Synthetic Data in Training"
_ICLR.cc/2024/Conference — ICLR 2024 Conference Withdrawn Submission_

### Official Review · Reviewer_VMMc · 2023-10-29

**Soundness:** 3 good
**Presentation:** 3 good
**Contribution:** 3 good
**Rating:** 5
**Confidence:** 4

**Summary:**

The paper explores different aspect into using synthetic data for human detection in UAV images. This includes the interplay between the quantity of real vs. synthetic images, quantifying the distribution gap and how this is related to performance, and the impact of different testing data presenting domain shifts. The domain gap of a trained detector is measured by summing the Mahalanobis distances between in- and out-of-domain data as represented by the penultimate layer of the detector. For training with synthetic data, the PTL scheme is used, where various degrees of synthetic data are sampled based on there domain similarity with the real data.

**Strengths:**

+ Comprehensive analysis on the studied task
+ Reveals some interesting findings related to the use of synthetic data

More details in questions/comments below.

**Weaknesses:**

- As the study is limited to one task, it is not clear if results will generalize
- There are only one synthetic dataset tested, and only one synthetic to real transformation

More details in questions/comments below.

**Questions:**

* While the study is comprehensive and highlights some interesting aspects related to the use of synthetic data, it is also limited to one application and one type of synthetic data. It is not clear that the same conclusions would hold for different tasks and datasets.

* It could also be a different situation if the synthetic data was generated differently (for example using computer graphics vs. generative deep learning), or if the sim2real operator was different (for example something more state-of-the-art than a vanilla CycleGAN).

* One aspect related to bridging the gap between synthetic and real data is domain randomization, which is not discussed in relation to the experiments. There could be benefits in using synthetic data with a large degree of domain gap compared to real data, but where the complexity can promote optimization and generalization of the trained model. Here, a discussion around the differences in regards of sample quality/fidelity vs. diversity would be of interest.

* Is the proposed measure of domain difference used in synthetic data selection for PTL? This is not clear from the explanations. If so, how does it compare to the original method used? And if not, would it be beneficial to use the proposed measure?

* The findings formulated in the introduction, and the conclusions discussed in Section 6, are not very clear, and the potential impact is limited. It is not clear how these would translate to different datasets and synthetic data generation methods.

---

### Official Review · Reviewer_Eyf8 · 2023-10-29

**Soundness:** 2 fair
**Presentation:** 3 good
**Contribution:** 3 good
**Rating:** 3
**Confidence:** 3

**Summary:**

This work explored how synthetic data affects machine learning model. They measured the distribution gap via Mahalanobis distance in feature space. They investigated about (1) the effect of synthetic data in terms of distribution gap, (2) the importance of real data, (3) the importance of sim2real transformation quality, and (4) the characteristics of the synthetic data pool.

**Strengths:**

- Synthetic data plays an important role in the field where data acquisition is difficult. For example, synthetic data is mainly used in optical flow [A]. Since synthetic data is widely used, research is needed to understand how it is used. This work conducted it.
- Based on Sim2Real transformation, they conducted the experiments of synthetic data in terms of the size of the real dataset, the size of the synthetic dataset, data selection methods (random, PTL), and synthetic pool comparison.

[A] FlowNet: Learning Optical Flow with Convolutional Networks, ICCV 2015

[B] Fake it till you make it: face analysis in the wild using synthetic data alone, ICCV, 2021

**Weaknesses:**

- The authors expanded the domain gap used in PTL to the dataset comparison. The expansion is incremental.
- In Sec. 5.1, the authors commented, “…, particularly the impact of synthetic images more significant when the amount of real data is small”. However, in Fig. 2(a), |Performance of VisDrone w/ synth - Performance of VisDrone w/o synth| is larger in 200 samples.
- In Table 2, the more real data authors use the smaller the domain gap. For example, the distribution gap is changed to 35.1 (20) → 31.7 (50) → 23.2 (100) → 19.8 (200) in the 50% column of Okutama. Why is VisDrone used more, and has the distribution gap decreased?
- The authors commented, “random selection is generally more effective in reducing the distribution gap using synthetic data than PTL”. It seems that the measure of the distribution gap used is not sufficient to delve into Sim2Real transformation alone. To delve into the synthetic data, it is better to provide the various views using another distribution measure, the number of parameters used in learning, etc.
- PTL shows a higher performance than random selection despite the high distribution gap, which means the model is biased toward the reference dataset. The authors said the quality of Sim2Real transformation is important. What is the quality authors said?

**Questions:**

Please refer to the weaknesses.

---

### Official Review · Reviewer_Jce6 · 2023-10-31

**Soundness:** 3 good
**Presentation:** 4 excellent
**Contribution:** 2 fair
**Rating:** 5
**Confidence:** 4

**Summary:**

The paper provides an analysis of using synthetic data along with real data to train a model, focusing on two scenarios: one where both training and test data come from the same domain (same-domain task), and another where they come from different domains (cross-domain task). It proposes an in-depth analysis of the domain gap present in the sim2real transformations by measuring the normalized sum of the Mahalanobis distances from the training set for each test data. The authors build on top of prior work (PTL) as their setup to perform the analysis. PTL is a framework that expands the training set by adding virtual images using a GAN to apply visual transformations (PLT -- Progressive Transformation Learning for Leveraging Virtual Images in Training). The paper shows the results of varying the amounts of synthetic data and the performance on five datasets.

**Strengths:**

- This paper is well-written and easy to follow. The motivation of the proposed work is compelling, given the significance of using synthetic data to train a model that can be used in real-set scenarios.

- The supplementary materials provide an extended set of experimental results varying the number of sim + real data, along with the implementation details, which adds clarity and reproducibility.

**Weaknesses:**

- Limited analysis: as shown in [1], the Mahalanobis distance is useful for comparing a single synthetic data point against the distribution of real data. However, prior work has shown good metrics on how similar the representations learned from different datasets are [2], including sim2real distribution shifts [3].

- Limited contribution: the proposed work heavily relies on [1], and only proposes to train the sim2real transformer by initializing from the model learned in the previous PTL iteration.

- It is hard to measure the contribution of the proposed analysis and insights without comparing the proposed PTL and established domain adaptation techniques.

[1] Shen, Yi-Ting, et al. "Progressive Transformation Learning for Leveraging Virtual Images in Training." Proceedings of the IEEE/CVF Conference on Computer Vision and Pattern Recognition. 2023.

[2] Kornblith, Simon, et al. "Similarity of neural network representations revisited." International conference on machine learning. PMLR, 2019.

[3] Mishra, Samarth, et al. "Task2sim: Towards effective pre-training and transfer from synthetic data." Proceedings of the IEEE/CVF Conference on Computer Vision and Pattern Recognition. 2022.

**Questions:**

- Is it plausible to affirm that improving the sim2real transformation quality is more important than reducing the distribution without considering domain adaptation techniques to bridge the gap between datasets?

---

### Official Review · Reviewer_Gemq · 2023-11-01

**Soundness:** 2 fair
**Presentation:** 2 fair
**Contribution:** 2 fair
**Rating:** 5
**Confidence:** 3

**Summary:**

This paper is focused on maximizing the impact of synthetic data during training. It explores the sim2real transformation and discusses three key factors that influence its quality: real data, the selection of synthetic data, and the synthetic data pool. The study examines how these factors affect the effectiveness of synthetic data in improving model performance and enhancing domain generalization abilities. Additionally, the paper introduces a new evaluation metric designed to measure the distribution gap between two datasets. Using human detection in UAV-view images as the target task, the paper provides a lot of discussions and findings regarding the use of synthetic data in training.

**Strengths:**

+ Investigating the influential factors when utilizing synthetic data for training is both interesting and valuable for a wide range of tasks.

+ A multitude of experiments have been conducted to examine the impact of various factors on the effectiveness of using synthetic data in training for human detection in UAV-view images

+ The discussions and findings have the potential to offer valuable insights into optimizing the utilization of synthetic data during training

**Weaknesses:**

- It is unclear whether the discussions, conclusions, and findings of this paper can be generalized for other applications when using synthetic data in training.

   - The experiments conducted in this paper are focused on the task of human detection in UAV-view images, which raises questions about the applicability of the derived findings and conclusions to other tasks.
   - Given the limited scale of both real and synthetic data used in the experiments (in the thousands), it is worth considering whether the paper's findings hold true when applied to large-scale synthetic training data.


- The paper does not mention or utilize FID (Fréchet Inception Distance), a commonly used metric for measuring domain gap. It would be beneficial to mention why FID is not employed in this paper and to discuss the advantages of the proposed metric over FID.

-  Some of the findings in this paper have been demonstrated in existing work on Sim2REAL

**Questions:**

See Weaknesses